# Development of Absorbent Using Amylose-Graphite Composite Electrode for Removal of Heavy Metals

**DOI:** 10.3390/membranes11120930

**Published:** 2021-11-26

**Authors:** Shuang Li, Guizani Mokhtar, Ryusei Ito, Toshikazu Kawaguchi

**Affiliations:** 1Graduate School of Global Food Resources, Hokkaido University, Sapporo 060-0809, Japan; alicelishuang@eis.hokudai.ac.jp; 2Faculty of Engineering, Hokkaido University, Sapporo 060-0809, Japan; g_mokh@yahoo.fr (G.M.); ryuusei@eng.hokudai.ac.jp (R.I.)

**Keywords:** absorbent, heavy metal, amylose, graphite, absorption isotherm

## Abstract

Amylose of *Phragmites Australis* captures heavy metals in a box consisting of sugar chains. However, its absorption rate is low in the period of the month scale. Therefore, the electrochemical driving force was used to promote the absorption rate in this research. Amylose was doped with TiO_2_ porous graphite electrode. The composted absorbent was characterized using XRD(X-ray diffraction), SEM (Scanning Electrode Microscopy), Raman spectroscopy, and electrochemical methods. The affinity and maximum absorption amount were calculated using the isotherm method. In this study, Pb^2+^, Cu^2+^, Cd^2+^, and Cr^6+^ were chosen to demonstrate because these heavy metals are significant pollutants in Japan’s surface water. It was found that the maximum absorption was Cu^2+^ (56.82-mg/L) > Pb^2+^ (55.89-mg/L) > Cr^6+^ (53.97-mg/L) > Cd^2+^ (52.83.68-mg/L) at −0.5 V vs. Ag/AgCl. This is approximately the same order as the hydration radius of heavy metals. In other words, the absorption amounts were determined by the size of heavy metal ions. Subsequently, the mixed heavy metal standard solution was tested; the maximum absorption amount was 21.46 ± 10.03 mg/L. It was inferred that the electrochemical driving force could be shown as the ion size effect in the mixed solution. Despite there being no support for this hypothesis at this time, this study succeeded in showing that the electrochemical driving force can improve the ability of the absorbent.

## 1. Introduction

Rising population, factory expansion, urbanization, agricultural activities, and various chemicals usage have led to serious environmental issues [1,2]. Water contamination is the most important of these environmental issues to examine, as water quality has a direct link to human health, as hazardous substances are absorbed in bodily organisms through the food chain and drinking water [1]. Copper, chromium, cadmium, and lead are the most common heavy metal contaminants in Japan. Heavy metals are highly toxic, non-biodegradable, and easily accumulate in human organs, causing various diseases and health disorders [3,4,5,6]. After the investigation, it was discovered that the high concentration of copper discharged into the river and infiltrated into the surface of the soil due to copper mining in the Ashio Copper Mine upstream of the Watarase River killed numerous crops and fish died downstream of the Watarase River in Japan in 1890 [7]. In 1905, the Japanese government decided to build sedimentation ponds to remove the mining poison. *Phragmites australis* was planted in large quantities surrounded Watarase River to absorb harmful heavy metals from the water [8]. *Phragmites australis* is still used today to protect water safety and wildlife [7]. Heavy metal contamination in water bodies has resulted in numerous illnesses, such as ITAI-ITAI disease, lead poisoning, and skin fester [9]. Thus, regarding these illnesses, WHO stipulates the standard for different types of heavy metals in water bodies: the copper concentration should be less than 2 mg/L, chromium is set at 0.05 mg/L, cadmium must be less than 0.003 mg/L, and lead must be maintained below 0.01 mg/L [10]. In addition to complying with regulations, water treatment technology must be implemented to achieve water recycling.

Coagulation, chemical precipitation, oxidation, reverse osmosis, membranes, solid-phase extraction, electrochemical, and adsorption are some of the wastewater treatment technologies developed over the last few decades [11,12,13]. These technologies have some limitations; for example, chemical precipitation generates sludge, which needs further treatment and will eventually increase the cost [14,15,16]. Adsorption is the most widely used and recommended method for water treatment due to the adsorbent materials being possible to extract from natural resources, having a large quantity, being easily degradable, having the benefit of being highly efficient, and being less likely to cause secondary pollution to the environment [17,18,19]. Many materials are used as adsorbent materials for water treatment, such as zeolites, resins, polymeric membranes, and clays that can adsorb heavy metals [20,21]. However, some adsorbent materials showed limited capacity due to adsorption limitations, therefore many researchers are looking for various potential methods such as a modification to enlarge the capacity site [22,23,24].

Starch is a polysaccharide substance found in enormous quantities in nature, that is inexpensive, biocompatible, renewable, and degradable [25,26]. Amylose and amylopectin are two forms of semi-crystalline granules found in starch [27]. Amylose is less polymerized and has a higher water solubility than amylopectin. The focus of this study is on amylose as an adsorbent material in polymeric membranes. However, existing natural starch is not easily soluble in water; thus, changing the chemical and physical properties has become an approach for its wide use [28]. Most studies have demonstrated that modifying starch could change its response to pH value and temperature limitation [29,30]. There is also the possibility of adding specific functional groups to starch, such as amino, hydroxyl, and carboxyl groups to improve the adsorption affinity for heavy metal ions in water treatment. Various studies have succeeded in replacing the hydroxyl group of starch with different groups, such as carboxyl and acetyl group insertion [31]. There are many approaches to chemically altering starch like acid hydrolysis, cross-linking, and oxidation to replace the hydroxyl groups [32,33]. However, the hydroxyl group on the backbone of starch is one of the beneficial functional groups for heavy metal adsorption, and modifying its chemical properties does not essentially increase its adsorption surface area.

Therefore, in this study, we propose a new concept in which we use amylose in starch as a basic absorbent, and then add carboxyl functional groups at the periphery interface to increase its absorption surface area using the electrochemical driving force. Carbon materials are well known for having carboxyl functional groups surrounded by the backbone, low cost, and environmental friendliness, so they have become one of the common adsorbent materials [34,35]. Activated carbon is used as one of the common adsorbent materials because of its high surface area. However, it gradually loses its adsorption efficiency due to the limited sorption site [36]. Gang X. et al. suggested that graphitic carbon nitride nanosheets can adsorb cationic and anionic ions in wastewater because of the absorbents’ special shape and semi-conductivity properties [37]. Many studies have shown that the carbon materials are used as adsorbent materials; the pH value in the water body needs to be changed in most cases because the carboxyl groups are easily affected by different pH values [38,39]. Therefore, an adsorbent material that is not affected by the pH but has high absorption ability is requested. 

We believe that a composite of graphite and starch amylose could be a new heavy metal absorption material in wastewater treatment. Different amylose concentrations were used to modify the graphite carbon plates to analyze the absorption efficiency hoping to find the optimized concentration of amylose usage. Then, the material was conducted with a negative charge to calculate the absorption efficiency and compared with the results of uncharged experiments to confirm if the efficiency was improved. We expected the new absorption material to absorb various heavy metal ions from contaminated water efficiently and quickly.

## 2. Materials and Methods

### 2.1. Materials

All reagents used in this study were analytical grade. Standard solutions of copper (II) chloride, lead (II) chloride, cadmium (II) chloride, chromium (VI) oxide, potassium chloride, and amylose of starch were purchased from Wako, Japan. Potassium chloride was added as an electrolyte when the electrochemical method was applied. Hitachi Chemical Co., Ltd. donated porous carbon plates. 

### 2.2. Instruments

In this study, the characterization of graphite porous carbon plates was measured using Scanning Electrode Microscopy (SEM, Hitachi High-Tech Corporation, S4800, Tokyo, Japan), X-ray diffraction (XRD, Rigaku Corporation, RINT2200V, Tokyo, Japan), Raman spectroscopy (JASCO Corporation, RMP500, Tokyo, Japan), and a digital multimeter (ADVANTEST, R6450, Tokyo, Japan). The amylose and graphite porous carbon plate was used in the vacuum system and new adsorbent materials were dried in an oven at 80 °C. The adsorption of heavy metals was performed using an Electrochemical Instrument (HOKUTO DENKO CORPORATION, HZ-7000, Tokyo, Japan), with three electrodes connected. The working electrode (WE), counter electrode (CE), and reference electrode (RE) were amylose/graphite porous carbon plate,/graphite porous carbon plate, and Ag/AgCl electrode, respectively. The concentration of heavy metal ions was measured using an Atomic Absorption Spectrophotometer (AAS, Hitachi High-Tech Corporation Z-2000, Tokyo, Japan).

### 2.3. Characterization of Graphite Porous Carbon Plate

Two types of graphite carbon plate could be a potential material. Two types of graphite porous carbon plates were prepared; SEM was used to observe the appropriate porous size, for which samples were controlled in magnification: 4000, working distance: 17.4 mm, acceleration voltage: 5 KV, and emission current: 10 μA condition. SEM images (Figure 1a,b) show the pore size of the two types of graphite porous carbon plate; the cavity of the pore was measured at 5 μm. According to the SEM images, two graphite porous carbon plate exposed a similar pore size, and the multilayer sheets were also expressed.

However, XRD was used to identify the carbon components and detect the carbon crystal phase of two porous carbon plates using Cu-Ka radiation of 1.5405 Å wavelength and a scanning speed of 2° min^−1^. The surface property of porous carbon electrodes was investigated using Raman spectroscopy (JASCO, RMP 500, Tokyo, Japan). In carbon materials, Raman spectroscopy is used to detect bond strengths via lattice vibration. Figure 1c,d shows the XRD and Raman spectra of two types of porous carbon plates. As a result, two types of porous carbon showed a typical graphite peak in XRD patterns at 2θ = 26.7° [40]. In Raman spectra, the main features of the G-band and D-band appear at 1582 cm^−1^ and 1350 cm^−1^, respectively. The Raman peak around 1582 cm^−1^ is known as the G-band typical Raman peak of bulk crystalline graphite. This peak is the basic vibration mode of graphite crystals. The size of the crystal influences the intensity [41]. The vibration of the crystalline carbon edge of the graphite, known as the D-band, produces a peak value of 1350 cm^−1^.

Regarding the Raman spectra, graphite-dominated porous carbon was noticed because the high G-band value was exposed in both carbon electrodes. The high value of the G-band and D-band ratio means good electroconductivity property [42]. The ratio of G-band and D-band of two carbon electrodes were 0.45 and 0.48, respectively. Furthermore, anatase and rutile composite crystal phases was observed in Figure 1d after being added to the carbon graphite plate with anatase peaking at 2θ values of 47.6°, 53.5°, and 55.1°; and 2θ values peak of rutile were found at 35.6° and 61.0°, which coincided with crystal planes of anatase (200), (105), and (211), and rutile crystal planes of (101) and (310) [43]. Graphite carbon plates have good electroconductivity, but their hydrophobic characteristic makes them unsuitable for water mediation. Thus, anatase and rutile were added to the graphite carbon plate to change the surface of the graphite carbon plate for a hydrophilic property. TiO_2_-doped graphite porous carbon plate was used as a potential material for water mediation in this study.

### 2.4. Doping Amylose into Graphite Porous Carbon Plates

The vacuum pump was used to vacuumize TiO_2_-doped graphite porous carbon plates, which were subsequently immersed in 10g of amylose solutions and vacuumed for 120 min. The weights of amylose/TiO_2_ doped porous carbon plates were measured after they were removed from the solution and dried in an oven at 80 °C for 120 min. Figure 2 shows the dried weight of amylose/TiO_2_-doped porous carbon increased tendency. It shows that the weight of TiO_2_-doped porous carbon plate became saturated in 120 min of doping time. The lowest carbon area (1.0 cm × 1.0 cm × 0.2 cm) showed 0.1 g change and the biggest carbon plate area (9.0 cm × 15.0 cm × 0.2 cm) showed 0.9 g change; thus, the amylose of starch increased proportionally with the carbon plate area. Since the electrochemical method was to be used, the area of carbon plate was chosen to be 5.0 cm × 9.0 cm × 0.2 cm for ease of operation. 

### 2.5. Absorption Isotherm

Absorption isotherms show how the absorbents interact with adsorbate and their distribution on the surface. It has also been used to compare the affinity of absorbents for the removing contaminants from the hydrosphere.

The Langmuir isotherm is the most common method for describing the absorption capacity, applied to real ions absorption processes in many studies. There are already several common isotherms to express the efficiency of adsorbents. However, in this study, amylose was doped into the graphite multilayer sheet plate; the common adsorption isotherm could not certainly be consistent in this study. Here, one new absorption isotherm was discussed based on the Langmuir sorption isotherm. It assumes that at first, adsorbates are uniformly distributed on a homogeneous monolayer absorbent. We also assumed that despite inside amylose being random distribution, the surface of amylose also shows a homogeneous monolayer. Therefore, the equation is given as follows:(1)$$HM^{2+}+A_{site}\;⇆HM^{2+}-A_{site}$$

[$HM^{2+}$] means the free concentration of heavy metal ions in solution, [$A_{site}$] means the amylose absorption site. [$HM^{2+}-A_{site}]$ exhibits the absorbed concentration of heavy metals in the new absorbent. Regarding the chemical equilibrium reaction, the affinity constant [*K*] can be written as follows:(2)$$K=\frac{\left[HM^{2+}-A_{site}\right]}{\left[HM^{2+}\right]\left[A_{site}\right]}\;$$

The calculation of the concentration of [$A_{site}$] can simply be considered the maximum absorption ability of [$A_{site}$] subtraction.
(3)$$\left[A_{site}\right]={\left[HM^{2+}-A_{site}\right]}_{max}-\left[HM^{2+}-A_{site}\right]\;$$

Equation (4) is substituted from Equation (3), and the equation can be rewritten as follows:(4)$$\frac{1}{\left[HM^{2+}-A_{site}\right]}=\frac{1}{K{\left[HM^{2+}-A_{site}\right]}_{max}}\times \frac{1}{A_{site}}+\frac{1}{{\left[Heavy\;metal^{2+}-Amylose_{site}\right]}_{max}}$$

Here, *K* and [$A_{site}$]*_max_* exhibit affinity of absorbent and the maximum absorption capacity, respectively. The heavy metal absorption isotherm analysis using amylose doped into graphite porous carbon electrode was described.

## 3. Results

### 3.1. Heavy Metal Absorption Amount Analysis

This study on heavy metal ion absorption amount using amylose was conducted to verify heavy metal ion absorption ability. Then, the amylose/TiO_2_-doped graphite porous carbon was described at a given potential change for a 120 min heavy metal ion absorption process. Figure 3a shows the absorption amount of Pb^2+^ using amylose/TiO_2_-doped graphite porous carbon in 0.3 g at a −0.5 V charged potential applied. The absorption amount is 66.11 mg/L; approximately 26.4% of Pb^2+^ had been absorbed at the end of 120 min. However, at the initial concentration, 100 mg/L of Pb^2+^ exposed to approximately 47.52% of Pb^2+^ was absorbed in 120 min. According to the absorption rate for lead removal, 100 mg/L could be a critical initial concentration for the absorption equilibrium. Figure 3b shows the maximum absorption amount of 57.69 mg/L for 250 mg/L Cu^2+^ absorption using amylose/TiO_2_-doped graphite porous carbon in 0.3 g at a −0.5 V charged potential applied; however, the capacity of absorption in Cu^2+^ solution showed 40.76% at the initial 100 mg/Lof Cu^2+^. As a result, the absorption amount of the developed absorbent has the best absorption performance at 100 mg/L of Cu^2+^. Figure 3c shows the absorption amount of Cd^2+^, and the results show that even though absorption amount continued to grow as cadmium concentration increased, the absorption amount decreased over 200 mg/L of Cd^2+^ absorption, which is a similar phenomenon to Pb^2+^ and Cu^2+^. The maximum absorption amount and capacity were 47.81 mg/L and 42.65%, respectively. Cr^6+^ absorption amount is also described in Figure 3d, which shows the lowest absorption amount among these heavy metals. The maximum absorption amount value was shown 39.68 mg/L for an initial concentration of 150 mg/L Cr^6+^ absorption, and the maximum absorption capacity appeared on the 100 mg/L, which showed approximately 34.68% absorption capacity. The maximum absorption amount and capacity are summarized in Table 1. Amylose/TiO_2_-doped graphite porous carbon for removal of heavy metals exhibited the best absorption amount at the initial concentration of 100 mg/L individually, but it showed a different value of absorption amount and capacity. The size of the ionic radius of Pb^2+^, Cu^2+^, Cd^2+^, and Cr^6+^ is 1.75 Å, 1.28 Å, 1.49 Å, and 1.25 Å, respectively. The size of heavy metal ions determined the absorption ability and the absorption affinity.

### 3.2. Heavy Metal Absorption Isotherm of Amylose/TiO_2_ Doped Graphite Porous Carbon

In this study, amylose was fixed at a certain amount into TiO_2_-doped graphite porous carbon in each plate. To understand the absorption rate and how the electrochemical driving force method works for the removal of heavy metals, the affinity constant *K* and ${\left[HM^{2+}-A_{site}\right]}_{max}$ were calculated in Table 2 based on the established absorption isotherm. The new absorbent order of each heavy metal’s absorption ability is as follows: Cu^2+^ > Pb^2+^ > Cr^6+^ > Cd^2+^. The value of maximum absorption amount calculated from the absorption isotherms is similar to the results appeared in the experiments, the absorption amount of Cu^2+^ and Pb^2+^, and Cr^6+^ and Cd^2+^ were approximately at the same value. It is inferred that the ionic radius of Cu^2+^ and Pb^2+^ are close; Cr^6+^ and Cd^2+^ are at a similar value as well. Since the maximum absorption amount of the single component amylose absorbent showed that Pb^2+^(20.89-mg/L) > Cu^2+^ (19.15-mg/L) > Cd^2+^ (15.86-mg/L) > Cr^6+^ (13.29-mg L), the amylose-doped TiO_2_-doped graphite porous carbon absorbent had a higher absorption ability than the single component amylose absorbent.

Figure 4 showed that even if all of the heavy metal concentrations could not pass through the linear lines for every single point, most points are on the line. Therefore, the new absorbent could fix this new absorption isotherm. As previously mentioned, amylose does not show a homogeneous distribution on the surface of TiO_2_-doped graphite porous carbon. First, potential charges occurred with amylose covered by the carbon plate’s surface; subsequently, inside amylose acted as the charge potential continued.

### 3.3. Practical Application

As is well known, heavy metals contaminated with water have frequently co-existed with multi-species. The absorption of heavy metals in mixed water samples was investigated. Each heavy metal’s initial concentration ranged from 10 mg/L to 250 mg/L. The amount of absorbent was charged with the same potential as the individual experiment. Figure 5 shows each heavy metal absorption amount for 120 min absorption progress in a mixed solution. The absorption capacities for heavy metal removal are exhibited differently from a single ion absorption. The maximum absorption capacities occurred in the initial concentration of 100 mg/L of each ion, as with the individual heavy metal absorption results. The average of mixed heavy metal absorption was 21.46 mg/L. The maximum absorption amount value is shown in Table 3 for each heavy metal absorption based on the absorption isotherm. The order of maximum absorption for mixed heavy metals is Cd^2+^ > Cr^6+^ > Pb^2+^ > Cu^2+^. The maximum absorption amountsdeclined, and the order of the value expressed the opposite to the individual absorption value. However, this result can be considered to follow the ionic radius as well.

In the mixed solution, amylose on the surface absorbed the smaller size of ions making the inside of amylose unavailable for work, resulting in a two-fold decrease in average absorption levels. Despite an absorption competition in mixed solutions, the absorption amount is still as good as expected.

## 4. Conclusions

The electrochemical driving force approach was used to test a new absorbent using an amylose/TiO_2_-doped graphite porous carbon electrode for heavy metal absorption in this study. The performance of amylose-doped TiO_2_-doped graphite porous carbon absorption depended on the initial concentration of heavy metals, and the absorption equilibrium concentration was 100 mg/L. It was shown that the absorption amount and ability were advanced for single component amylose absorbent and single heavy metal absorption. Meanwhile, the devised absorption isotherm is in accordance with the absorption trend of the new absorbent material for heavy metals absorption, and the ionic radius determined the order of heavy metal maximum absorption. There is an opposite order compared to one heavy metal ion and mixed solution. In contrast, the electrochemical driving fore method for compositing porous carbon plate and amylose could drastically improve the absorbent ability compared to bulk amylose application.

## Figures and Tables

**Figure 1 membranes-11-00930-f001:**
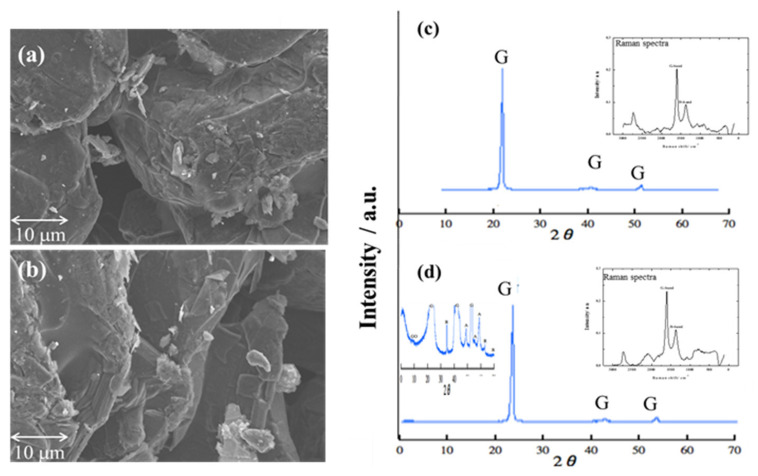
SEM imagines of (**a**) porous carbon; (**b**) TiO_2_-doped graphite porous carbon. (I: 10 μA V:5 kV Mag: 4000 WD: 17.4 mm). XRD and Raman spectra of (**c**) porous carbon; (**d**) TiO_2_-doped graphite porous carbon.

**Figure 2 membranes-11-00930-f002:**
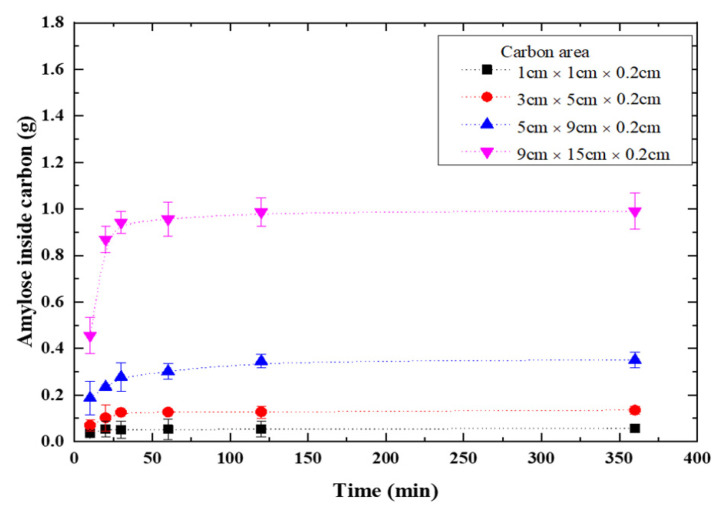
Time dependence of amylose doped into TiO_2_-doped graphite porous carbon.

**Figure 3 membranes-11-00930-f003:**
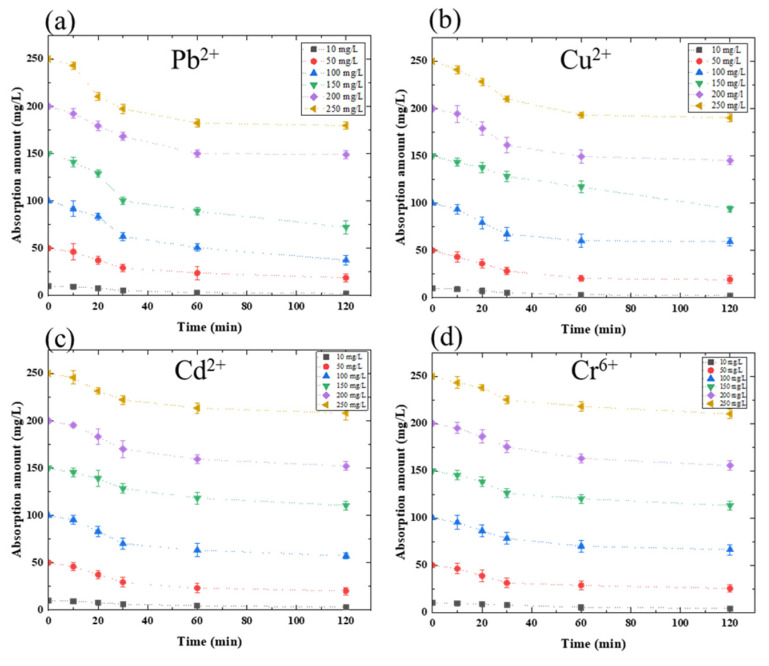
Absorption amount in different heavy metal concentration solutions: (**a**) Pb^2+^; (**b**) Cu^2+^; (**c**) Cd^2+^; (**d**) Cr^6+^ using 0.3 g amylose/TiO_2_-doped graphite porous carbon at −0.5 V potential.

**Figure 4 membranes-11-00930-f004:**
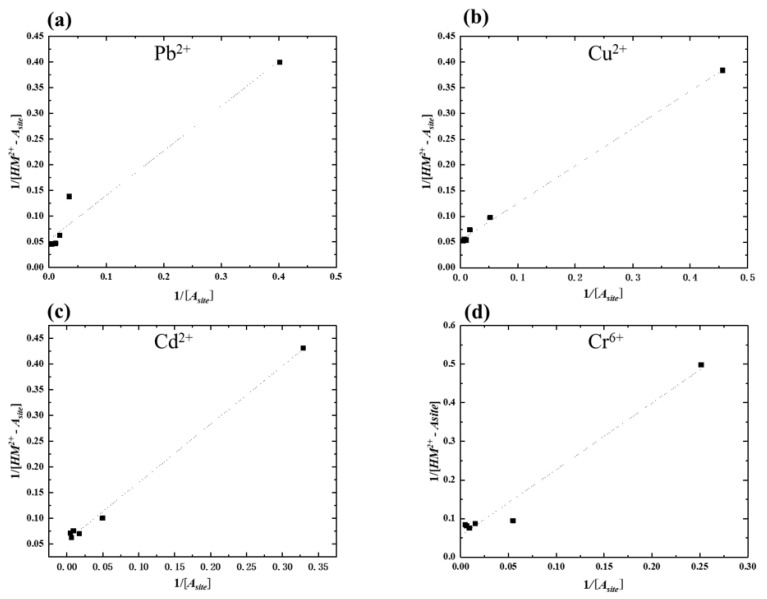
The absorption isotherm fitting for each heavy metal absorption: (**a**) Pb^2+^; (**b**) Cu^2+^; (**c**) Cd^2+^; (**d**) Cr^6+^ using 0.3 g amylose/TiO_2_-doped graphite porous carbon at −0.5 V potential.

**Figure 5 membranes-11-00930-f005:**
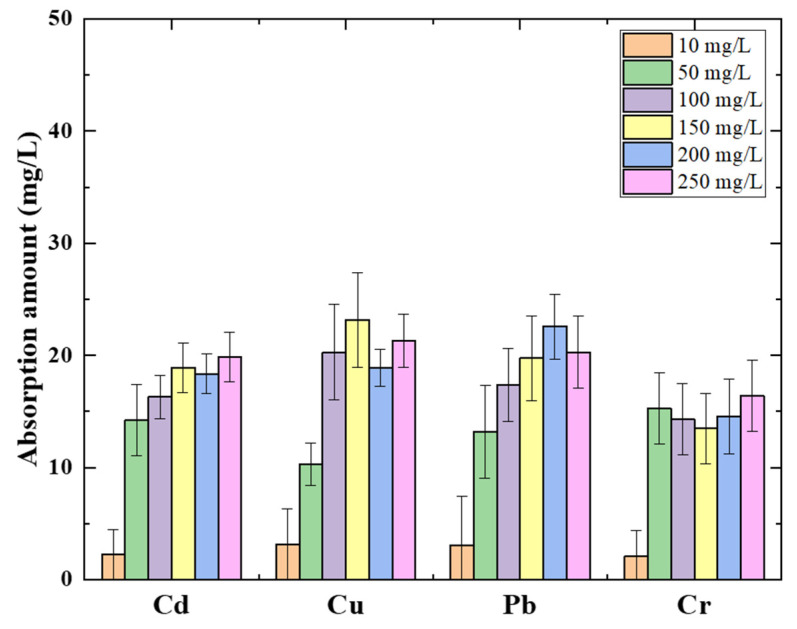
Absorption amount in different commixture of heavy metals concentration solutions using 0.3 g amylose/TiO_2_-graphite porous carbon at −0.5 V potential.

**Table 1 membranes-11-00930-t001:** The absorption amount and capacity for each heavy metal removal by using 0.3 g amylose/TiO_2_-doped graphite porous carbon at −0.5 V potential.

	Pb^2+^	Cu^2+^	Cd^2+^	Cr^6+^
Absorption amount (mg L^−1^)	66.11	57.69	47.81	39.68
Maximum absorption capacity (%)	47.52	40.76	42.65	34.68

**Table 2 membranes-11-00930-t002:** The maximum absorption amount and affinity for each heavy metal absorption by using 0.3 g amylose/TiO_2_-doped graphite porous carbon at −0.5 V potential.

	Pb^2+^	Cu^2+^	Cd^2+^	Cr^6+^
Maximum absorption amount (mg/L)	55.89	56.82	52.83	53.97
*K*	0.06	0.07	0.05	0.03

**Table 3 membranes-11-00930-t003:** The maximum absorption amount and affinity for mixed heavy metal absorption using 0.3 g amylose/TiO_2_-graphite porous carbon at −0.5 V potential.

	Pb^2+^	Cu^2+^	Cd^2+^	Cr^6+^
Maximum absorption amount (mg/L)	30.21	27.75	44.31	32.31
*K*	0.017	0.019	0.007	0.009

## Data Availability

Not applicable.

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
