# Peer review of "Development of Absorbent Using Amylose-Graphite Composite Electrode for Removal of Heavy Metals"

_membranes, 2021, doi:10.3390/membranes11120930_

Round 1
Reviewer 1 Report
- Reference 3 not cited in the MS
- It is a common practice to include highlights of the work. The authors should consider including this in the manuscript.
- Criteria for selection of the carriers should be established.
- Authors should explain why it is suitable for this method.
- Is there any scale-up or any geometrical similarities that can be established from this work that can be used for scaling up process?
- Fig.4 Missing in the main text.
- Last line in conclusion is too general and was not discussed in the manuscript. Should remove or reword.
- No information on the characterization of the wastewater
Author Response
Dear review editor,
Thank you for your precious suggestion. We revised the manuscript based on your advice. Your advice helps the manuscript be more easily understanding. We responded to your point one by one. Please check the files and the manuscript.
Best regards,
Shuang Li

Reviewer 2 Report
The authors presented the amylose-graphite composite electrode for elimination of heavy metals. The adsorbent was characterized using XRD, SEM, Raman spectroscopy and electrochemical methods. The adsorption ability towards Pb(II), Cu(II), Cd(II), and Cr(VI) was shown.
The paper topic is interesting but the paper is very difficult to read. The description of the figures, parameters needs corrections. The explanation is not deep enough and are unclear. The authors presented different values in Tables and Figures but the description is the same (figure captions, table captions). Lack of details description results that Figures 4 seems to be a repetition. The methodology is not described in details. My recommendation reject.
Detailed comments are set out below:
- In the keywords removal, composite should be placed. Moreover, absorption or adsorption isotherm?
- Lines 122 and 126 - 10 µA or 10 mA?
- Figure 1 captions needs corrections. Please describe a, b, c, and d properly. The quality of fig. 1 c and d should be improved.
The I, V, Mag. should be explained in the paper e.g. ……..in magnification (Mag): 4000, working distance (WD): 17.4 mm, acceleration voltage (V): 5 KV, and emission current (I): 10 µA condition.
- Figure 2 caption should be corrected…. doped into TiO2 Please checked also other figure captions.
- Line 167 – Absorption isotherm is in the title of the chapter whereas below Adsorption. Please correct.
- Line 171-179 should be rewritten.
- Number of Equations should be right-aligned.
- The adsorption amount should be defined.
- Adsorption or absorption??????Is it the same for authors? because both are used interchangeably.
- The parameters included in Table 1 should be explained – the proper equations should be added.
- Where is Fig. 4?
- What was the difference between parameters presented in Table 1 and 2, 3?different values, the same parameters. Please specify.
- Figures 3 looks as a repetition of previously presented data (Fig. 3).
- References should be unify.
Please see the attachment.

Author Response
Dear review editor,
Thank you for your precious suggestion. We revised the manuscript based on your advice. Your advice helps the manuscript be more easily understanding. We responded to your point one by one. Please check the following answers and the manuscript.
Best regards,
Shuang Li

Reviewer 3 Report
Kindly add the following recent published article to your literature survey in the introduction part
Performance evaluation of modified fabricated cotton membrane for oil/water separation and heavy metal ions removal
In the material part add more details about the materials used including their chemical and physical properties
There is a large interference between the results and experimental within the article
In the material and method part you should add only materials, preparation methods and characterization techniques without any results of analysis or characterization
the discussion of results obtained should be transported to the results and discussion part.
So, Figure 1 and 2 and the discussion relevant to them should be transported to the results part.
Add FTIR and SEM as a characterization method for Doping Amylose into Graphite Porous Carbon Plates
The expression of adsorption capacity in (mg of heavy metal /g of adsorbent) is more efficient you can read it in the above paper
The adsorption experiment should be mentioned in detail
Author Response

(The authors gave the same response as above.)

Round 2
Reviewer 1 Report
Thanks for answered my question and now the MS has been improved to satisfy the international readers.
Author Response
Dear review editor,
Thank you very much for reviewing our manuscript again in your busy schedule. According to your previous last valuable comments, our paper has been greatly improved. We appreciate a lot for your precious comments, it also helped us with the next manuscript preparation.
Thank you very much.
Best regards,
Shuang Li

Reviewer 2 Report
The paper was improved according to the reviewer remarks and now is better quality.
Additional corrections are needed which can be done at the proofreading stage:
- Figure 1 caption still needs corrections, ";", "." should be checked again and indicate which one is the X-ray diffraction and Raman spectra in the figure.
- Please see the response to point 5 and 9 in your replay. Please unify.
Author Response
Dear review editor,
Thank you very much for reviewing our manuscript again in your busy schedule. According to your previous last valuable comments, our paper has been greatly improved. We revised our manuscript again based on your comments again. The specific revised parts have been marked in the manuscript. Please check it.
We are looking forward to your more response.
Best regards,
Shuang Li

Reviewer 3 Report
The authors didnt add the the chemical and physical properties of the matrials used ( heavy metal salts )
the authors didn't response well to this comment :
In the material and method part you should add only materials, preparation methods and
characterization techniques without any results of analysis or characterization. the discussion of results
obtained should be transported to the results and discussion part. he discussion of results obtained
should be transported to the results and discussion part. So, Figure 1 and 2 and the discussion relevant
to them should be transported to the results part.
this article which is so close to your work didn't cited in your work (Performance evaluation of modified fabricated cotton membrane for oil/water separation and heavy metal ions removal )
Author Response
Dear review editor,
Thank you very much for reviewing our manuscript again in your busy schedule. According to your previous last valuable comments, our paper has been greatly improved. Moreover, we have to apologize for the last improper response. We revised our manuscript again based on your comments again. The specific revised parts have been marked in the manuscript. Please check it.
We are looking forward to your more response.
Best regards,
Shuang Li
